# Are 2D shallow water solvers fast enough for early flood warning? A comparative assessment on the 2021 Ahr valley flood event

Shahin Khosh Bin Ghomash[1], Heiko Apel[1], and Daniel Caviedes-Voullième[2,3,4,5]

[1]Section Hydrology, GFZ German Research Centre for Geoscience, Potsdam (Germany)
[2]Simulation and Data Lab Terrestrial Systems, Jülich Supercomputing Centre, Forschungszentrum Jülich (Germany)
[3]Institute of Bio- and Geosciences: Agrosphere (IBG-3), Forschungszentrum Jülich (Germany)
[4]HPSC TerrSys, Geoverbund ABC-J (Germany)
[5]Centre for Advanced Simulation and Analytics, Forschungszentrum Jülich (Germany)

**Correspondence:** Daniel Caviedes-Voullième (d.caviedes.voullieme@fz-juelich.de)

**Abstract.** Flash floods pose a distinct challenge compared to traditional fluvial flooding, with infrastructure-based solutions proving less effective. Effective responses hinge on advanced early warning systems providing actionable information, emphasising the necessity for computational flood forecasting models. However, hydrodynamic models, renowned for accuracy and completeness, face limitations due to computational intensity.

This study explores two 2D flood forecasting models, RIM2D and SERGHEI, both with GPU implementations which allow to maximise the forecast lead time. While RIM2D is less computationally intensive, suitable for operational use, SERGHEI, with higher computational costs, targets large-scale High-Performance Computing (HPC) systems.

The assessment of applicability and trade-offs is carried out on the 2021 Eifel flood event, particularly in the lower Ahr valley. A set of simulations were performed at various resolutions from 1m to 10m, which reveal similar accuracy among both models at coarser resolutions, yet discrepancies arise at finer resolutions due to the distinct formulations. Both models exhibit a rapid computational cost escalation, but at resolutions equal to or coarser than 5m, forecasts are remarkably faster than real-time—ideal for operational use, paving the way for their use in early warning systems. However, higher resolutions necessitate multi-GPU and HPC capabilities, underlining the importance of embracing such technology in addressing broader flood domains.

## 1 Introduction

The accurate prediction and timely communication of future natural disasters, particularly floods, have become crucial components for disaster management strategies. Early warning systems play a key role in reducing the loss of life and property during such events, allowing appropriate preventive measures to be taken before hand (Šakić Trogrlić et al., 2022). One of the key tools in these systems is computational hydrodynamic models enabling the simulation and forecasting of flooding in response to varying conditions.

2D SWE models have been around for quite some time and have been implemented in multiple use cases e.g. (Pasculli et al., 2021). Two-dimensional Shallow Water equation (2D SWE) solvers have a long history in flood modelling (De Almeida and

Bates, 2013; Hill et al., 2023), and are a promising approach for enhancing the accuracy and efficiency of early warning systems (Apel et al., 2022; Cea and Costabile, 2022; Costabile et al., 2023). However, until recently, the practical application of 2D SWE models in early warning systems has been very limited due to various challenges related to computational capabilities, data assimilation, and real-time decision-making.

The 2021 flooding event in the Ahr Valley (Germany), stands as a stark reminder of the destructive power that extreme weather events can unleash. In July 2021, the region experienced a catastrophic flood event, resulting in loss of life, displacement of residents, and extensive damage to infrastructure, homes, and landscapes (Mohr et al., 2022). Out of the 184 fatalities in Germany, 133 occurred along the river Ahr – a Rhine tributary. The relatively small size of the Ahr River basin ($\sim 900 \, \text{km}^2$) and its morphological features including narrow streams in gorges, result in a stream network with limited capacity for handling sudden influxes of water which consequently makes many areas in the Ahr prone to flash floods. Flash floods are characterised by their sudden onset and fast escalation (Kelsch, 2001).

Catastrophic events such as the Ahr floods are rare and have mostly a local effect, which partially explains why they have received historically less attention than large river floods and likely remain under-represented (Paprotny et al., 2018). However, climate change is likely to make such events more frequent and more intense (Donat et al., 2016; Myhre et al., 2019), thus arguably making them more prominent even in regions in which they have been atypical. From a prevention point of view, regions potentially strongly affected by flash flood events can have very little room for structural improvement. This is the case of the Ahr valley, with urbanised areas occupying the very narrow floodplain, and surrounded by steep valleys. With limited potential for structural defences, early warning systems are the key tool to allow a continued safe inhabitation of these areas, so that both loss of life and economic damage may be minimised.

Early warning systems pose many challenges. The spatial and temporal scales of flash floods and the consequent short lead times, make it challenging to run timely and accurate flash flood simulations producing actionable information (Merz et al., 2020). The Eifel flash floods were a severe stress test for the existing early warning system, which resulted in short lead times, untimely warnings, incomplete/outdated/inaccurate information and inconsistent recommendations (Thieken et al., 2023b). The nature and timing of the issued flood warnings played a role in the scale of the casualties (Thieken et al., 2023a). Thieken et al. (2023a) argues that warnings communicating rainfall amounts are far less interpretable (by the general population, but possibly also by managers and emergency responders) than water levels and inundated areas. However, forecasting water levels, inundated areas, flow velocities and time of arrival of a flash flood requires, firstly, a hydrodynamic extension of the existing flood forecasts, which are based on hydrological model output at selected rive gauge locations, and secondly, a high level of sophistication in the hydrodynamic flood model employed.

This means that an appropriately high resolution model is mandatory to capture the complex geometries of valleys, streams and urban areas in order to reliably predict inundation areas and water levels. Second, the nature and complexity of the physical phenomena does not allow for 1D simplifications, far more commonly implemented (Hill et al., 2023) than 2D models. Finally, the simulation needs to be computed fast enough to allow for sufficient lead time. Until recently, this was not achievable and remains the main impediment to wide spread adoption of 2D models in flood modelling practice (Hill et al., 2023). However, as 2D SWE solvers enhance to more effectively leverage high-performance computing (HPC), new possibilities for early

warning with 2D SWE models arise. In general terms HPC has enabled physics-based geoscientific modelling to achieve unprecedented detail (Alexander et al., 2020), and in particular, shallow water solvers are now fully exploiting this with the use of GPU computing (Morales-Hernández et al., 2020), as well as leveraging into massively parallel super-computing (Caviedes-Voullième et al., 2023; Morales-Hernández et al., 2021).

The question that naturally follows is: can HPC-enable shallow water solvers achieve sufficient accuracy and lead time to improve early flood warning systems in order to better manage events such as the Ahr valley floods? And, does this technology translate into better and more actionable information? We explore these questions using two surface flow solvers, namely the RIM2D and SERGHEI solvers, using different mathematical models and HPC implementations to assess not only the feasibility but the trade-offs.

## 2 Methods

### 2.1 Numerical models

We use two 2D surface flow solvers in this work, namely SERGHEI (Caviedes-Voullième et al., 2023) which solves the fully dynamic shallow water equations, and RIM2D (Apel et al., 2022) which solves a local inertia approximation. The key advantage of SERGHEI is that it can be deployed on very large scale HPC systems, leveraging on massively parallel scientific hardware. This allows to offset the comparatively larger computational cost of solving the full shallow water equations. In contrast, RIM2D allows to solve the comparatively cheaper local inertia equations, arguably requiring fewer computational resources, albeit in the current version v0.2 limited to a single GPU.

### 2.1.1 Full shallow water solver: SERGHEI

SERGHEI (Caviedes-Voullième et al., 2023) solves the fully dynamic shallow water equations

$$\frac{\partial \mathbf{U}}{\partial t} + \frac{\partial \mathbf{F}}{\partial x} + \frac{\partial \mathbf{G}}{\partial y} = \mathbf{S_b} + \mathbf{S_f},$$

$$\mathbf{U} = \begin{bmatrix} h \\ q_x \\ q_y \end{bmatrix} \quad \mathbf{F} = \begin{bmatrix} q_x \\ \dfrac{q_x^2}{h} + \dfrac{1}{2}gh^2 \\ \dfrac{q_x q_y}{h} \end{bmatrix} \quad \mathbf{G} = \begin{bmatrix} q_y \\ \dfrac{q_x q_y}{h} \\ \dfrac{q_y^2}{h} + \dfrac{1}{2}gh^2 \end{bmatrix},$$

$$\mathbf{S_b} = \begin{bmatrix} 0 \\ -gh\dfrac{\partial z}{\partial x} \\ -gh\dfrac{\partial z}{\partial y} \end{bmatrix} \quad \mathbf{S_f} = \begin{bmatrix} 0 \\ -\sigma_x \\ -\sigma_y \end{bmatrix}. \tag{1}$$

where the conserved variables $\mathbf{U}$ are water depth $h$ $[L]$ and momentum components $[L^2/T]$ in the Cartesian directions $q_x$ and $q_y$. $\mathbf{F}$ and $\mathbf{G}$ represent the fluxes. $\mathbf{S_b}$ is the bed source term, where $z$ is bed elevation $[L]$. $\mathbf{S_f}$ is the friction source term, where $\sigma_x$ and $\sigma_y$ are the friction slopes, here computed using Manning's equation. Finally, $g$ is gravitational acceleration $[L/T^2]$.

SERGHEI is written in C++ with hybrid parallelisation, i.e., MPI for distributed computations and Kokkos for shared memory computations. Kokkos (Trott et al., 2021) is a performance portability layer enabling it to reach both CPU and GPU backends. Consequently, SERGHEI can run on multiple GPUs, and is enabled for large scaling use in large HPC systems.

### 2.1.2 Local inertia solver: RIM2D

RIM2D is a 2D raster-based hydrodynamic model developed by the Section Hydrology of the German Research Centre for Geosciences (GFZ) in Potsdam, Germany. RIM2D solves the local inertia approximation to the Shallow Water equations (Bates et al., 2010), which has been widely shown to perform well for fluvial floodplain inundation applications e.g. (Falter et al., 2014; Neal et al., 2011; Apel et al., 2022). The local inertia approximation neglects the convective acceleration terms, and as a consequence decouples the fluxes in x- and y-direction. Thus, the fluxes $\mathbf{F}$ and $\mathbf{G}$ in (2) reduce to

$$\mathbf{F} = \begin{bmatrix} q_x \\ \frac{1}{2}gh^2 \\ 0 \end{bmatrix} \quad \mathbf{G} = \begin{bmatrix} q_y \\ 0 \\ \frac{1}{2}gh^2 \end{bmatrix}, \tag{2}$$

Conceptually, the local inertia formulation offers a more precise portrayal of the issue compared to the other simplified version of the SWE equations such as the zero-inertia (diffusive wave) model (De Almeida and Bates, 2013; Caviedes-Voullième et al., 2020). This is because, in contrast to the zero-inertia form, it keeps the local acceleration terms. In the discrete context, this implies that the fluid's momentum in a specific time step informs the subsequent step, thus imprinting a local acceleration in time. Thus, in describing shallow water flows physically, the local inertial formulation stands intermediary between the diffusion wave approximation and the comprehensive full-dynamic equations. While the original numerical solution offered by (Bates et al., 2010) is susceptible to instabilities under near-critical to super-critical flow conditions and for small grid cell sizes (De Almeida and Bates, 2013), the numerical diffusion proposed by de Almeida et al. (2012) has been additionally implemented in RIM2D.

RIM2D is written in Fortran, and ported to GPUs via CUDA Fortran libraries. It's worth noting that presently, RIM2D solely supports computations on a single GPU. However, efforts are underway to incorporate multi-GPU computing capabilities into RIM2D in the near future.

### 2.2 Study Case

The Ahr river is an 86 km long tributary of the Rhine river, located in the states of Rhineland-Palatinate and North Rhine - Westphalia (Germany), in the Eifel region. Our study domain focuses on the downstream reach of the Ahr river, spanning approximately 30km between the towns of Altenahr and Sinzig. In the first third of the reach the river valley is still very enclosed, but opens upstream to the town Bad Neuenahr-Ahrweiler into a wider valley floor. The area consists of mostly rural

areas, with a handful of small settlements, and the comparatively larger urban area of Bad Neuenahr-Ahrweiler (population of approximately 26500) (Truedinger et al., 2023). The average annual precipitation level of the region is below the German mean, at around 675 mm (Truedinger et al., 2023).

The nearly stationary low pressure system *Bernd* resulted in heavy rainfall events in Western and Central Europe in mid-July 2021 which triggered severe and sudden flooding especially in Belgium, the Netherlands, and Germany (Schäfer et al., 2021). The Ahr valley was one of the locations in Germany which was severely affected with accounting for an overall 70 percent of all fatalities in Germany (Truedinger et al., 2023), 189 in the area around the Eifel, making it the second largest water-related disaster in recent history in Germany (Thieken et al., 2023b) in terms of casualties. Numerous factors contributed to this extreme impact. Firstly, the Eifel embodies a low mountain terrain characterised by steep slopes and narrow valleys, extensively settled and cultivated by communities across an extended period. Consequently, the limited space results in a concentration of both population and structures in vulnerable zones. Furthermore, such areas are inherently susceptible to significant issues like mass movement, rapid erosive discharge, and substantial debris accumulation. These conditions notably caused extensive blockages, resulting in the destruction of numerous bridges along the Ahr river in July 2021, exacerbating the flood surge (Truedinger et al., 2023). During the 14 July 2021 event, water levels in the Ahr reached their highest values at the available gauging stations since the beginning of their measurements. Although the exact water levels are unknown, as most gauging stations along the Ahr River were damaged or destroyed during the event, there are estimates of water levels of around 9 m at the Altenahr gauge (Mohr et al., 2022), where the normal water depths of the Ahr are less then 1 m.

## 2.3 Data and model setup

### 2.3.1 Spatial data

Three digital elevation model (DEMs) products provided by the German Federal Agency for Cartography and Geodesy (BKG) were used for model setup. The datasets DGM1, DGM5 and DGM10 with grid resolutions of 1, 5 and 10 meters were available. DGM5 and DGM10 are available finished products published by BKG. The DEMs were directly employed as the foundation for the simulations without undergoing any additional alterations nor crafting solutions for potential artifacts or lack of features. This is intentional, so that the simulations only rely on readily available datasets. Consequently, the simulations fail to realistically depict the river bed, instead portraying the average water surface in the Ahr river, which usually measures less than 1 meter (Apel et al., 2022). This approach is justified because both models used in this study operate based on water levels as boundary conditions rather than water depths and discharge. As a result, even with the presumed bed elevation, the water levels at the model boundary will consistently remain accurate, ensuring overbank flow and floodplain inundation happen in the correct locations and at the appropriate times. The buildings in the simulation domain were cut out from all three DEMs on the basis of building shape files provided by OpenStreetMap. An example of this can be seen in Figure 1 (white colouring in the lower figure). Consequently, building surfaces acted as closed reflective boundaries in the simulations.

Roughness manning values were assigned to the domain based on the 2020 Germany land cover classification derived from Sentinel-2 data (Riembauer et al., 2021). The data basis for the classification are atmospherically corrected Sentinel-2 satellite

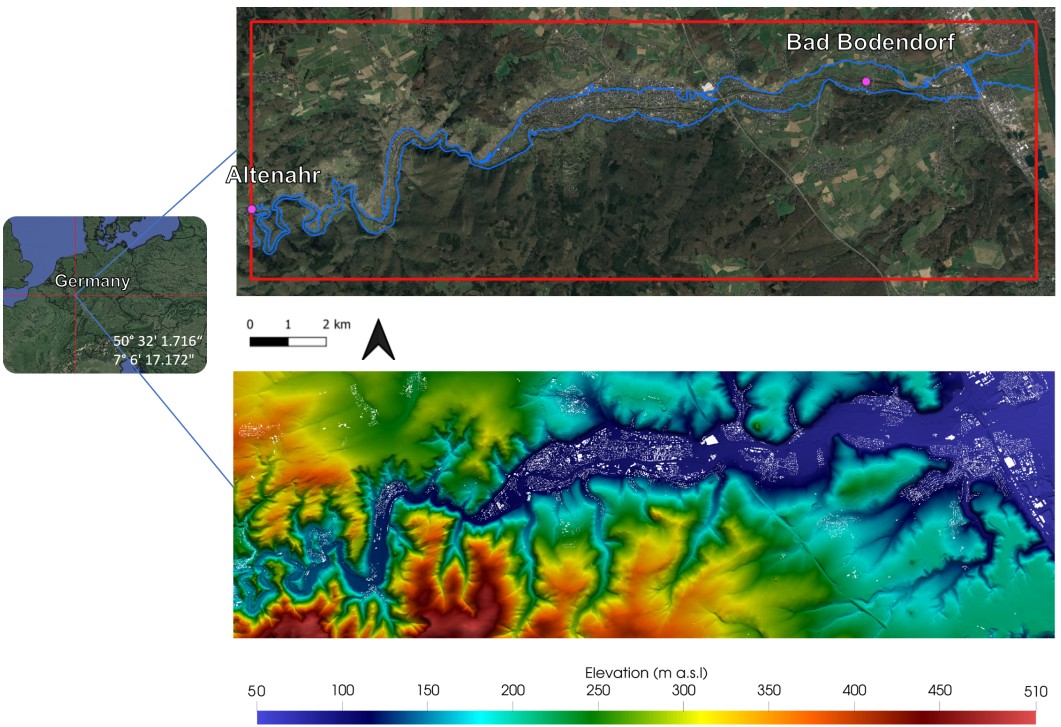

**Figure 1.** The red line delineates the boundary of the simulation domain, while the lower figure depicts the topography. In the upper figure, purple points indicate the positions of the Altenahr and Bad Bodendorf gauge stations. The blue line represents the maximum observed flood extent during the flooding event in 2021. Satellite imagery: © Google Earth 2024.

data (with the MAJA algorithm; data provided by EOC Geoservice of the German Aerospace Centre – DLR) as well as training data from reference data (e.g. OpenStreetMap) and the Sentinel-2 scenes themselves. This land cover was chosen for this study due to its relatively high grid resolution (10 meters). In addition to the mapped land use classes, the main Ahr river channel was added as an additional land category. Based on literature review, an appropriate Manning roughness value was chosen and

assigned to each land cover class in the simulation domain.Table 1 shows the Manning's roughness values assigned and the percentage coverage of each land cover type in the simulation domain.

The simulation exercise is performed intentionally in a blind fashion, without calibrating parameters such as roughness coefficients. The rationale for this choice is that the objective is to evaluate how feasible the use of these solvers is for early warning, and it cannot be assumed that a comprehensive calibration exercise would be available for every valley that the early

warning system oversees. Consequently, a blind approach based on available spatial data and standard parametrisations would be the only choice. Of course calibration would be desirable, but with the typical absence of calibration data (flood mapping) and the occasional need for a quick model setup in an operational case, an uncalibrated model is rather the standard use case in reality. Therefore we present the uncalibrated simulation results and do not dive into an in-depth model calibration in this study.

| Land Category | Manning roughness coefficient $[m^{-1/3}s]$ | Coverage [%] |
|---|---|---|
| Forest | 0.043 | 52.17 |
| Vegetation | 0.034 | 18.82 |
| Built-up/Sealed Areas | 0.027 | 11.37 |
| Bare Soil | 0.030 | 4.82 |
| Agriculture | 0.100 | 11.86 |
| River channel | 0.027 | 0.44 |
| Water bodies | 0.050 | 0.52 |

**Table 1.** Land cover categories, their respective area fraction in the domain, and their corresponding Manning's roughness values.

### 2.3.2 Flood event data for the inflow boundary

Inflow to the models is provided by the (official) reconstructed water levels (in metres above sea level) at the Altenahr gauge provided by the flood warning centre Rhineland-Palatinate (Mohr et al., 2022). The reconstruction is needed because the gauge was destroyed during the 2021 event. For model setup, observed water levels are assigned to the inflow cells in the domain. These cells are chosen on the river channel on the west boundary of the domain. In order to consider over-bank flow, cells neighbouring the river channel and with elevations below the maximum water level of the flood hydrograph were additionally selected. Water depths are assigned to the selected cells only when the river water levels exceed the cell elevation.

It is relevant to point out that a stage hydrograph was selected as an upstream boundary because this is what works natively best with RIM2D. Consequently, for comparability, the same boundary was used in SERGHEI, although SERGHEI can handle an inflow hydrograph.

### 2.3.3 Observation data for validation

For validation purposes in this study we rely on (i) the documented maximum flood extent provided by the State Agency for the Environment (LfU - Landesamt für Umwelt) of Rhineland-Palatinate, against which we evaluate the model skill in terms of flood extent; (ii) reconstructed stage hydrograph at Bad Bodendorf (Mohr et al., 2022), against which we compare the arrival time of the flood wave; (iii) and water depths derived from 65 high water marks reported by residents (Apel et al., 2022), from which water depths were derived to compare against simulated maximum water depths.

### 2.4 Inundation Performance Metrics

To quantitatively evaluate flood inundation in a domain, a diverse set of metrics are used to identify over- and under-predictions and their proportions. To compute these metrics, the maximum inundation maps of the simulations are evaluated against each other and the observed flood extent. At first, cells are classified with respect to Table 2. This is done by comparing the simulation results of RIM2D to SERGHEI. In addition, the results of each model is also compared to the observed inundation extent. From each comparison a confusion map is generated. From this map, the total counts of the indices shown in table 1 are computed

and used to calculate the domain-wide inundation metrics shown in Table 3. These metrics are adapted from Wing et al. (2017) and Bernini and Franchini (2013). It is important to note that when contrasting RIM2D with SERGHEI, the outcomes generated by RIM2D are considered as observed results, as indicated in Table 2

|  |  | Simulated | |
|---|---|---|---|
|  |  | Wet | Dry |
| Observed | Wet | True Positive (TP) | False Negative (FN) |
|  | Dry | False Positive (FP) | True Negative (TN) |

**Table 2.** Inundation confusion matrix. Each cell in the domain for a given simulation is compared to the corresponding cell in the observed grid and classified according to this table.

| Metric | Equation | Poor | Perfect | Description |
|---|---|---|---|---|
| Critical Success Index | $\dfrac{TP}{TP+FP+FN}$ | 0 | 1 | ratio of accurate wet cells to total wet cells and missed wet cells |
| Hit Rate | $\dfrac{TP}{TP+FN}$ | 0 | 1 | portion of observed wet cells reproduced by the model |
| False Alarms | $\dfrac{FP}{TP+FP}$ | 1 | 0 | portion of modelled wet cells which are erroneous |
| Error Bias | $\dfrac{FP}{FN}$ | 0 or inf | 1 | ratio of over-predictions to under-predictions |
| Bias Percentage Indicator | $100\left(\dfrac{TP+FP}{TP+FN}-1\right)$ | -100 or 100 | 0 | relative percentage error of the final extent of the flooded area |

**Table 3.** Flood inundation performance metrics

## 3 Results and discussion

### 3.1 Computational performance and runtime

One of the core questions of this study is whether these solvers are fast enough for their use in early warning systems. Consequently, we first examine the runtime and computational resources required to perform these simulations.

All simulations reported here were computed on NVIDIA A100 GPUs on the JUWELS Booster supercomputer at the Jülich Supercomputing Centre, as well as in the GFZ Linux Cluster.

Figure 2 shows the absolute (2a) and relative simulation runtimes for RIM2D and SERGHEI across the four resolutions (relative to each other in 2b, and relative to the event duration 2c) . Notably, at coarser resolutions (dx = 5 and 10 m), both models result in very short runtimes, clocking in at least 99 times faster than the duration of the 2021 flood event. This level of efficiency renders both models highly suitable for enhancing existing operational flood forecast systems while maintaining exceptional forecast lead times. Consequently, this capability facilitates detailed flood impact forecasting and swift responses.

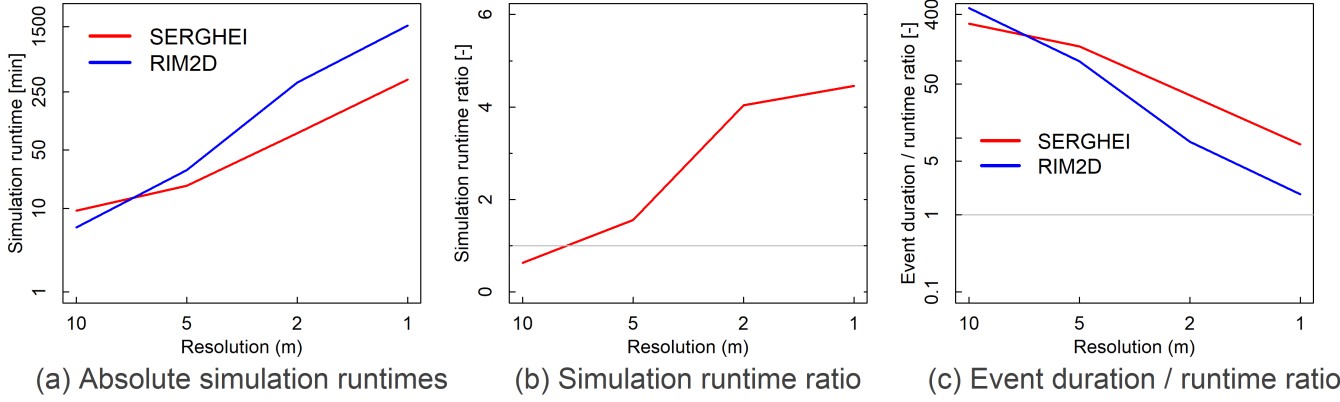

**Figure 2.** The absolute simulation run-times (a), the ratio between the simulation run-times of RIM2D to SERGHEI (b) and the ratio of the 2021 event duration to the simulation runtimes (c) for the dx = 10, 5, 2 and 1m simulations.

As resolutions become finer, the differences in runtime between the two models become more apparent. SERGHEI, employing multiple GPUs, results in runtimes up to 6 times faster than RIM2D, which in the current version 0.2 relies on a single GPU. At finer resolutions, i.e. large number of grid cells to be computed, the computational requirements surpass the parallel computing capabilities of a single scientific-grade GPU, necessitating multi-GPU implementations and some HPC capabilities for operational deployment. It is also notable that at the $dx = 10\,m$ resolution RIM2D does exhibit a slightly faster runtime compared to SERGHEI. This can be attributed to its less computationally intensive formulation, additionally indicating one GPU being adequate for simulations at that resolution.

In terms of the usability of these models for flood early-warning, as Figure 2c shows that all simulations were faster than the duration of the event. However, this ratio of event duration to runtime varies between 1 and 400 (for RIM2D) and 10 and 300 (for SERGHEI), depending on the resolution. It is also worthy to note that the dx = 10, 5, 2 and 1 m resolution models each consist of 1.3, 5.5, 34.7 and 139 million cells respectively.

It is relevant to highlight that no specific performance optimisation of the models was carried out for this particular case. Such optimisations could include compiler flags and hardware-based optimisations, domain decomposition strategies, and so on. These can potentially can reduce runtimes even further, but they are not necessarily generalisable across cases, software stacks and hardware. Consequently they are not particularly relevant for the objectives of this study. Nevertheless, such optimisation would be required for operational purposes, which would potentially boost performance even further.

Moreover, continued development in the implementation of the solvers will increase computational efficiency (e.g., by implementing a multi-GPU solver for RIM2D, or by dynamic balancing the load across GPUs), so this performance is expected to improve.

 ## 3.2  Flood model skill

The flood indicators illustrating the accuracy of both RIM2D and SERGHEI in replicating flooded areas across various simulations are depicted in Figure 3. Overall, both models demonstrate commendable performance, achieving high scores across all indicators. Notably, they exhibit relatively similar performance at coarser resolutions, but differences become more pronounced at finer resolutions. For instance, when considering the Critical Success Index (CSI), both SERGHEI and RIM2D yield comparable results with CSI values above 0.94 at $dx = 5$ and 10 m resolutions, whereas at finer resolutions (dx = 1 and 2 m), the CSI values drop into the eighties, highlighting more discernible disparities.

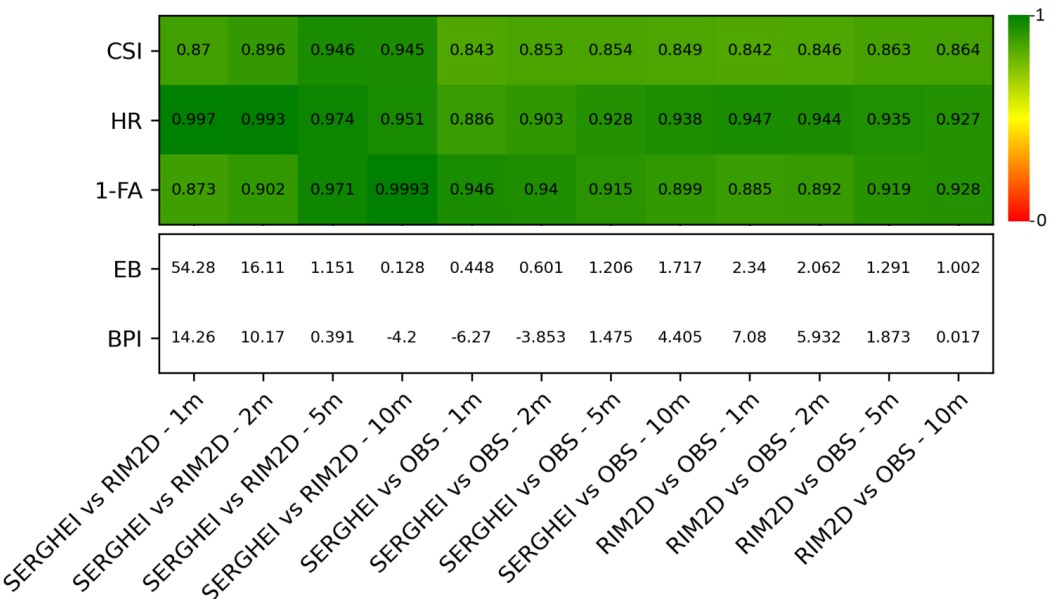

**Figure 3.** Comparison of flooded areas with the indices Critical Success Index, Hit Rate, False Alarm, and Error Bias

These variations at finer resolutions are evident in the Error Bias (EB) indicator as well. Specifically, in the 1 m and 2 m simulations, the scores for the two models diverge significantly, registering low scores of 54.28 and 16.11, respectively. Notably, the Hite Rate (HR) indicator stands out as an exception, with scores improving with better resolutions. This disparity is primarily attributable to SERGHEI depicting larger flooded areas in the finer simulations compared to RIM2D, resulting in a lower False Negative (FN) value (as indicated in Table 2) and consequently leading to a higher HR score.

## 3.3  Maximum flood depth

Figure 4 shows the difference in maximum depth between both models, for all four resolutions. Areas in which only one of the models predicts wet areas are categorised. It is important to recall that this is not the difference of water depths at any particular time, but the difference in the maximum depths reached during the entire event (which may be predicted at different

time by each solver, see subsection 3.4). The comparisons behave differently along the valley and are strongly affected by resolution. The narrower valley upstream of Mayschoss has reaches with very large differences in water depth, with SERGHEI predicting water depths up to 2.4m higher than RIM2D at $dx = 10$m, and up to 4m with $dx = 1$m. Near Rech there is a trend of SERGHEI predicting much lower water depths than RIM2D, with larger discrepancies at coarser resolutions. Conversely, upstream of Dernau SERGHEI again predicts higher water depths than RIM2D, but the differences are much smaller, in the order of $\sim 0.4$m to $\sim 1.4$m depending on the resolution. The differences in this narrow river valley with high water depths and flow velocities in the simulated flood events are likely caused by the different mathematical foundation of the models. Under these flow conditions the neglected convective acceleration in RIM2D might play a substantial role in the flow dynamics.

Consequently, in Bad Neuenahr-Ahrweiler, where the valley widens and the water depths and flow velocities reduce, the differences are significantly smaller, with a mix of positive and negative differences. In the region around and downstream of Bad Bodendorf SERGHEI tends to predict shallower depths than RIM2D. Additionally, at higher resolution there are more areas which are flooded by SERGHEI than RIM2D than at coarser resolutions. Of particular interest is that going from 5m to 2m generates additionally flooded areas by SERGHEI in Ahrweiler.

## 3.4 Time to maximum depth (lag)

Figure 5 shows the difference in time to maximum water depth (henceforth *lag* for brevity) between SERGHEI and RIM2D for the different spatial resolutions used, and Figure 6 shows the probability density functions of the lag. The lag is computed as follows: for both solvers, the time at which a particular cell reaches the maximum depth during the simulation is registered, and afterwards the difference (lag) between the time obtained by SERGHEI and RIM2D is computed.

There are both positive (RIM2D predicts earlier maximum depths) and negative (SERGHEI predicts earlier maximum depths) lags. Overall, negative lags only occur upstream of Mayschoss, in the narrowest part of the river valley. Clearly, the lag mostly increases from upstream to downstream (i.e., delays accumulate downstream). There are some local regions in which this does not hold (e.g., with $dx = 5m$, between Mayschoss and Rech). The second point is that the lag range reduces with increasing resolution. At 10 m resolution the lags are significant, up to $\sim 4\,h$, roughly $8\%$ of the duration of the event. At 1m resolution the lag drops to maximums of $\sim 2\,h$, roughly $2\%$ of the event duration.

In the reconstructed water level graph derived from the Bad Bodendorf gauge (Mohr et al., 2022), the highest water level occurs at 27.75 hours after the start of the simulation period (July 14, 2021), which is 2.5 h after the peak in the inflow hydrograph at Altenahr. Herein we refer to the time difference between the peak at this two stations as *hydrograph lag*, and we use this 2.5 h value as a reference. We computed the same hydrograph lag between both points for the simulations, and report it in Table 4. We also compute the difference between the simulated hydrograph lag and the 2.5 hour hydrograph lag estimated by the reconstructed hydrographs. Finally, this difference is expressed as an error relative to the reference hydrograph lag.

Table 4 shows that the hydrograph lag in SERGHEI reduces significantly with increased resolution, whereas the RIM2D hydrograph lag is far less sensitive. For SERGHEI, the lag difference is always positive, i.e., the peak at Bad Bodendorf is simulated later than the reference in SERGHEI. For RIM2D it is the opposite, it is always negative, meaning that RIM2D simulates a faster peak at Bad Bodendorf than the reference. The relative error is rather constant across resolutions for RIM2D,

| Solver | Hydrograph lag [h] | | | | Lag difference [h] | | | | Lag error [%] | | | |
|---|---|---|---|---|---|---|---|---|---|---|---|---|
| | 10m | 5m | 2m | 1m | 10m | 5m | 2m | 1m | 10m | 5m | 2m | 1m |
| SERGHEI | 4.50 | 3.75 | 3.25 | 2.75 | 2.00 | 1.25 | 0.75 | 0.25 | 80 | 50 | 30 | 10 |
| RIM2D | 1.61 | 1.52 | 1.40 | 1.46 | -0.89 | -0.98 | -1.10 | -1.04 | -35 | -39 | -44 | -41 |

**Table 4.** Simulated hydrograph lag between the Altenahr and Bad-Bodendorf gauges, and the difference relative to the 2.5 hour lag estimated from the reconstructed hydrographs.

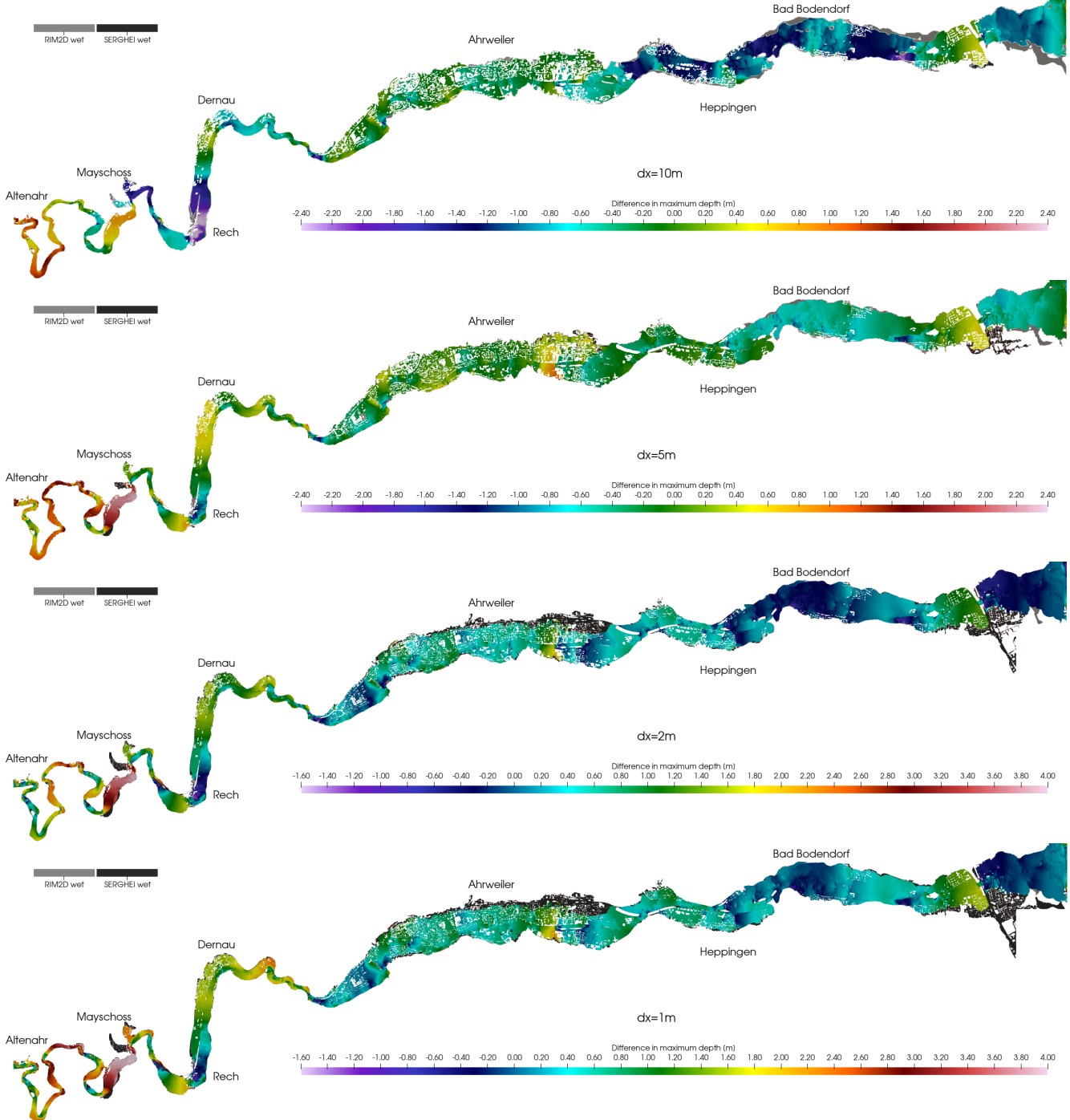

**Figure 4.** Difference in maximum water depth between the SERGHEI and RIM2D flood envelopes for all four resolutions. Positive values imply SERGHEI predicts higher maximum depths, and negative values imply RIM2D predicts higher maximum depths. The figure only compares true positive cells (flooded in both models). Gray colours show false positives and false negatives. Note the different ranges and colour scales for each spatial resolution.

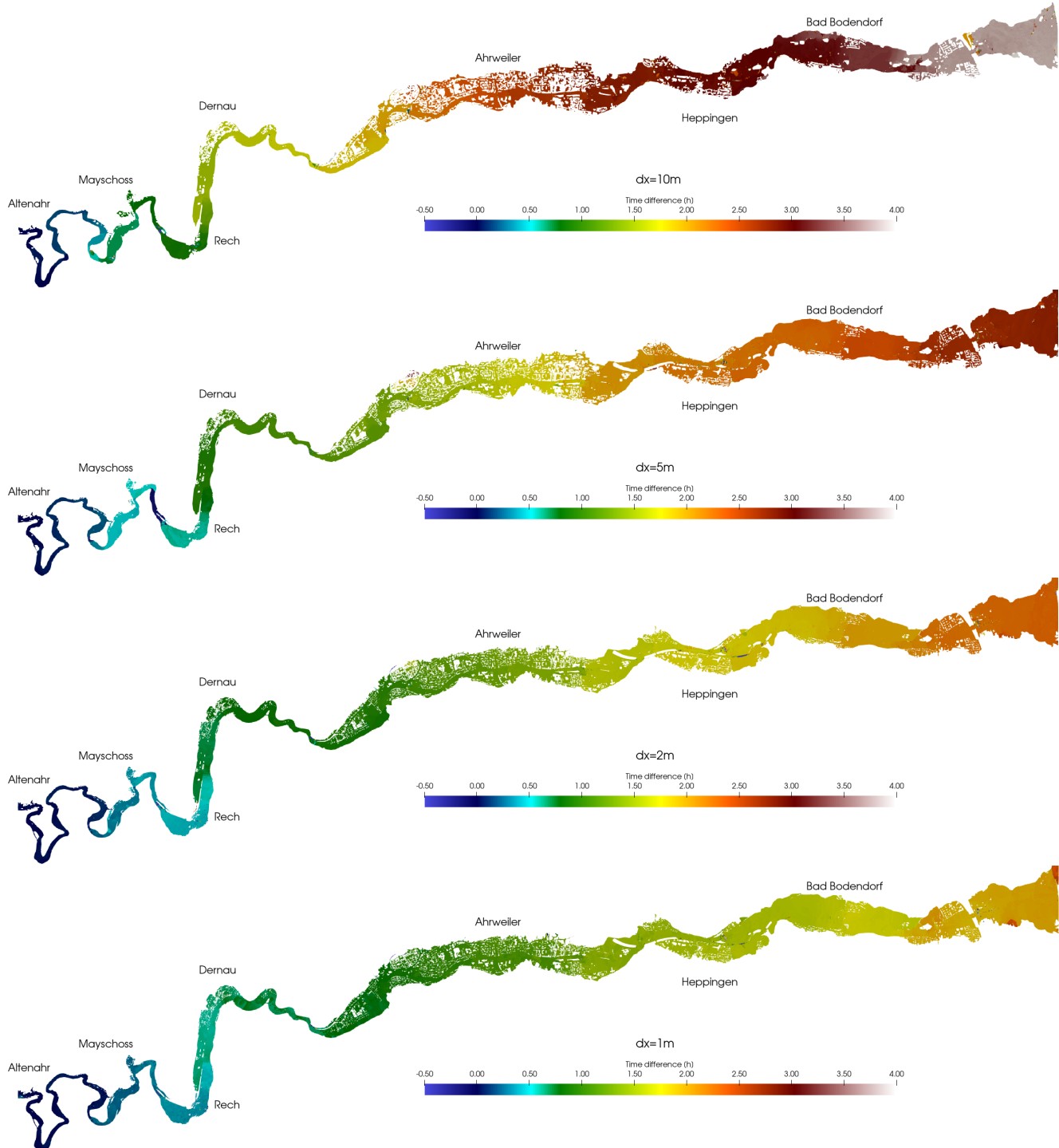

**Figure 5.** Difference in time to maximum water depth (lag) between the SERGHEI and RIM2D flood envelopes for all four resolutions. Negative values imply SERGHEI predicts earlier maximum depths, and positive values imply RIM2D predicts earlier maximum depths. The figure only compares true positive cells (flooded in both models).

around -40%, whereas for SERGHEI, as it is very sensitive to resolution, with very good results at high resolution, but rather poor results at 10m resolution.

These results suggest that the higher resolution SERGHEI simulations capture better the flood wave advancement, and decreasing resolution increasingly underestimates the flood wave movement. In contrast, RIM2D seems to overestimate the flood propagation speed, but is quite insensitive to resolution. It is worth mentioning that optimising each case individually through individual calibration would very likely lead to improved results, because simulated flow velocities and arrival times with different resolutions are sensitive to the roughness parameterisation (Bomers et al., 2019; Caviedes-Voullième et al., 2012; Ozdemir et al., 2013). Our results (together with contextual knowledge from the literature) also suggest that RIM2D may be more sensitive to roughness calibration (which is reasonable since the local-inertia simplifications give a somewhat higher weight to the friction model) and that it may be calibrated at a given resolution and results across resolution should improve. In contrast, whereas SERGHEI seems less affected by the lack of calibration, roughness parameters may need to be calibrated for each resolution.

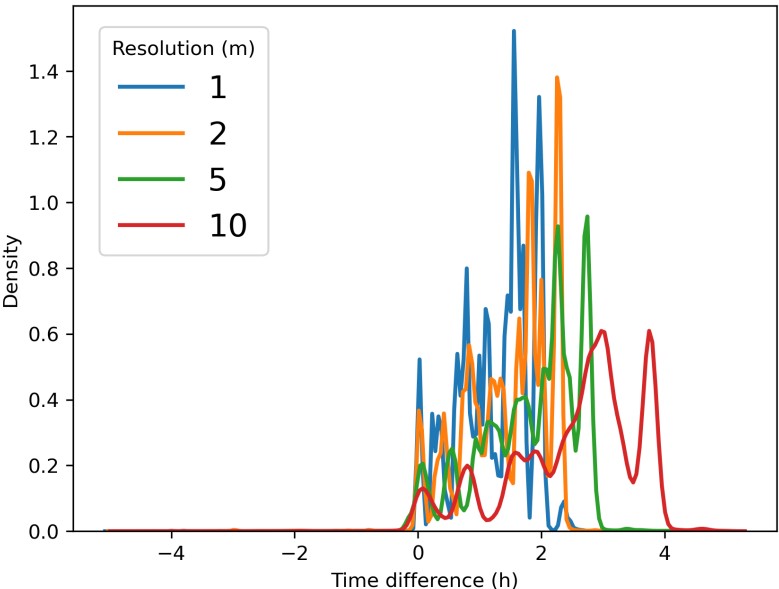

**Figure 6.** Probability density function of the difference in time to maximum water depth (lag) between the SERGHEI and RIM2D flood envelopes for all four resolutions for values. Negative lag values imply SERGHEI predicts earlier maximum depths, and positive lag values imply RIM2D predicts earlier maximum depths. The figure only compares true positive cells (flooded in both models). The lag range is limited to $[-5, 5]$ for readability.

To further explore the difference in the predicted lag beyond a single gauge point, Figure 6 shows the probability density function of the lag between the SERGHEI and RIM2D flood envelopes across all four resolutions. Negative values in the

graphs indicate that SERGHEI forecasts an earlier peak in maximum depths, while positive values mean that RIM2D predicts
an earlier peak. The comparison in the figure is limited to true positive cells (i.e., areas flooded in both models).

Broadly, the trend indicates that RIM2D consistently forecasts earlier maximum depths compared to SERGHEI across all
four resolutions (positive lag values), as already hinted by the lags at the Bad Bodendorf gauge point. The lag between RIM2D
and SERGHEI is more pronounced at coarser resolutions than at finer ones. As resolutions become finer, these disparities
diminish, and the differences tend to converge toward zero.

The comparisons shown in Figure 5, Figure 6 and with the reconstructed flood hydrograph at the Bad Bodendorf gauge im-
plies that RIM2D simulates faster flood wave propagation speed, which is insensitive to the model resolution. The insensitivity
to model resolution can be seen positively. The overestimation of the flood propagation speed, however, needs to be considered
when interpreting the results particularly in operational flood response, if this roughness parameterisation is used. SERGHEI
simulates a flood propagation in line with the reconstructed hydrograph at 1m resolution, and tends to underestimate it with
increasingly coarser resolutions. This again is also worth while considering when using the model at a particular resolution
with the presented roughness parameterisation for a particular purpose.

While studies like Martins et al. (2017) and De Almeida and Bates (2013) suggest that the local inertial approximation results
in slower flood propagation speeds compared to the full dynamic equations, it's important to note that Figure 6 solely depicts
the variance in time to reach maximum water depth which integrates additional processes and not only wave propagation
phenomena. Therefore, we argue that this lag disparity should not be construed as a metric for wave propagation. In the
evaluation of the flood propagation simulation it is also worth noticing that the reconstruction of the flood hydrograph at Bad
Bodendorf is also a hydrodynamic modelling result, thus also prone to errors in terms of water depths and timing and thus not
an absolute quantitative reference for the evaluation of the model results.

### 3.5   Comparison to maximum flood marks

Figure 7 shows a scatter plot comparison of field observations versus simulated maximum water depths at recorded post-flood
observations of maximum water marks at buildings. Figure 8 shows these same points explicitly in space, and color code the
relative difference between observations and simulations.

Figure 7 shows a slight general trend to under-predict rather than over-predict water depths across the domain. Most im-
portantly, it allows to see how the comparison shifts with resolution. $R^2$ values clearly overall deteriorate for the coarser
resolutions. SERGHEI trends towards underestimating more with coarser resolutions, but the opposite is the case for RIM2D.

RIM2D shows closer agreement with observed water marks with coarser resolution models. As resolution increases, the
discrepancy between simulated and observed depths becomes more pronounced. This is particularly clear from Figure 7,
where it is possible to see how for the 1m and 2m resolutions, RIM2D tends to under-predict maximum depths. At the 10m
resolution, there are a few RIM2D points which greatly over-estimate observations.

This discrepancy primarily stems from RIM2D's tendency to generate smaller flooded areas in higher resolution setups
compared to coarser ones, resulting in under-predicted depths or missing inundation in areas further from the main river
channel (see subsection 3.6 for details), which corresponds to the broad behaviour of the points in space (Figure 8).

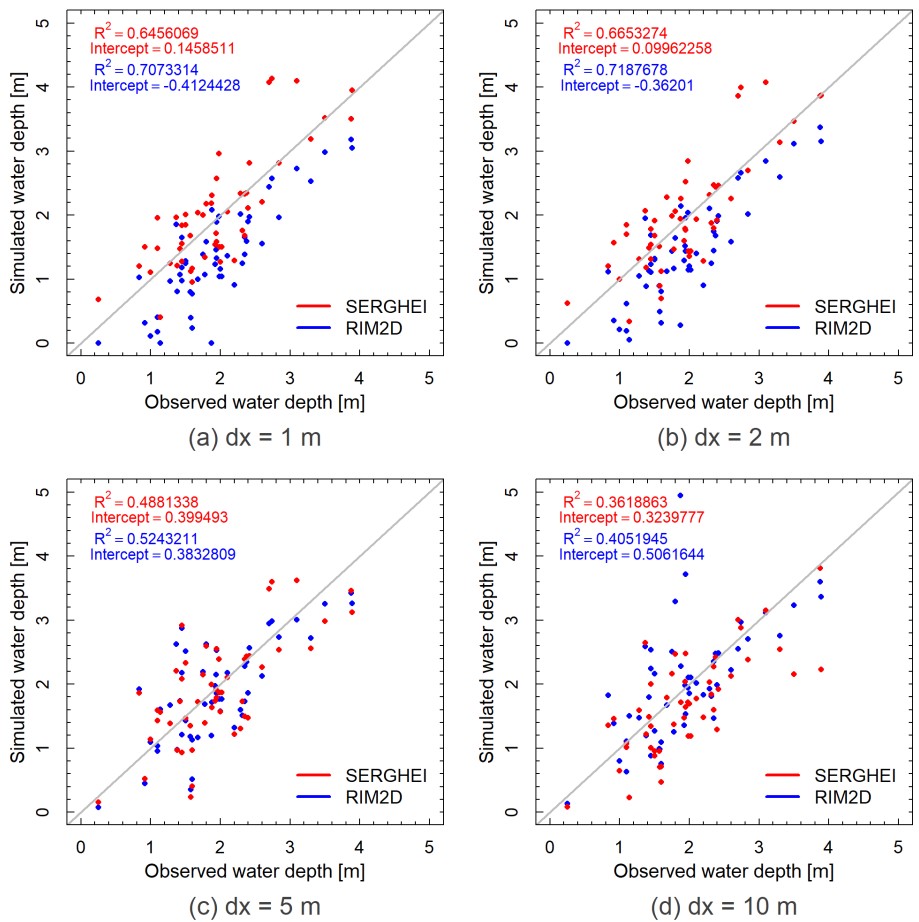

**Figure 7.** Scatterplot comparison of observed maximum water depths agains RIM2D and SERGHEI simulations for all four resolutions.

In contrast, for SERGHEI, a distinct trend among the four resolutions is not apparent, and all model configurations tend to produce deviations within a similar range, arguably with better estimations at higher resolution (Figure 7(a)). Some additional insights can be drawn by also accounting for the spatial distribution of the comparison points, as shown in figure Figure 8. There is a somewhat improving trend towards higher resolution, in which the points located farther from the main river channel which predominantly exhibit under-predicted water depths somewhat improve. In certain cases this is because the predicted inundated area falls short of the location of the points.

Close inspection of the location of the recorded water marks shows that many of the predicted points with lowest scores, especially at coarser resolution, are the result of poor representation of the buildings in the computational grid. This is illustrated in Figure 9, where it can be seen that for a coarse resolution (e.g., 10m) many of the observation points fall in cells which are identified as buildings, although the point itself is not in the building. As resolution increases more of these points fall into valid areas of the computational domain. This is likely to happen since these observation points are often water marks on walls

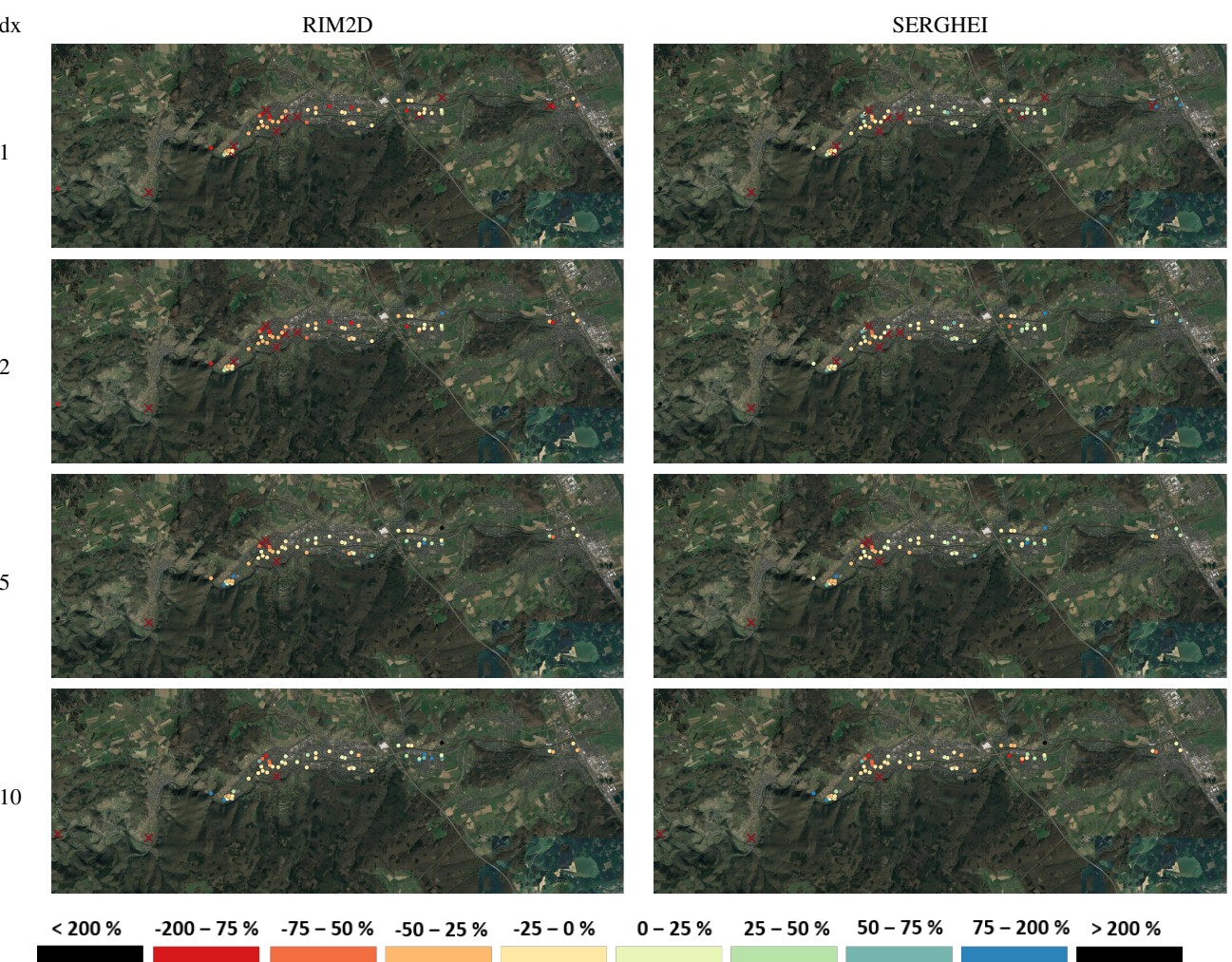

dx | RIM2D | SERGHEI

| < 200 % | -200 − 75 % | -75 − 50 % | -50 − 25 % | -25 − 0 % | 0 − 25 % | 25 − 50 % | 50 − 75 % | 75 − 200 % | > 200 % |

**Figure 8.** Error (as percentage) in the simulated water depth compared to observed for RIM2D and SERGHEI for all four resolutions. Positive values indicate an overestimation in the water depth and negative values show an underestimation. The × marks represent points which fall onto the building footprint rasterised onto the cartesian grid. Satellite imagery: © Google Earth 2024.

or urban furniture close to buildings. It is important to highlight that these building representation challenges are present in both the RIM2D and SERGHEI simulation scenarios. To offset this issue, we also allow a search for valid (non-building) cells adjacent to the cell containing the observed point. This allows some leeway to capture more points into the analysis. Moreover, aside from the issues relating to observed points, Figure 9 highlights the effect that resolution can have on properly capturing the complex urban environments, even in a fully inundated area as shown in this image. It highlights that although overall metrics may suggest that the 10m resolution is sufficient to broadly capture the flood, the inundation dynamics in complex

urban built-up areas are prone to errors with raster resolutions to coarse to match the urban complexity. In the presented case study in small towns and villages this appears to be the case at 10m resolution, but this might be different in other urban fabrics.

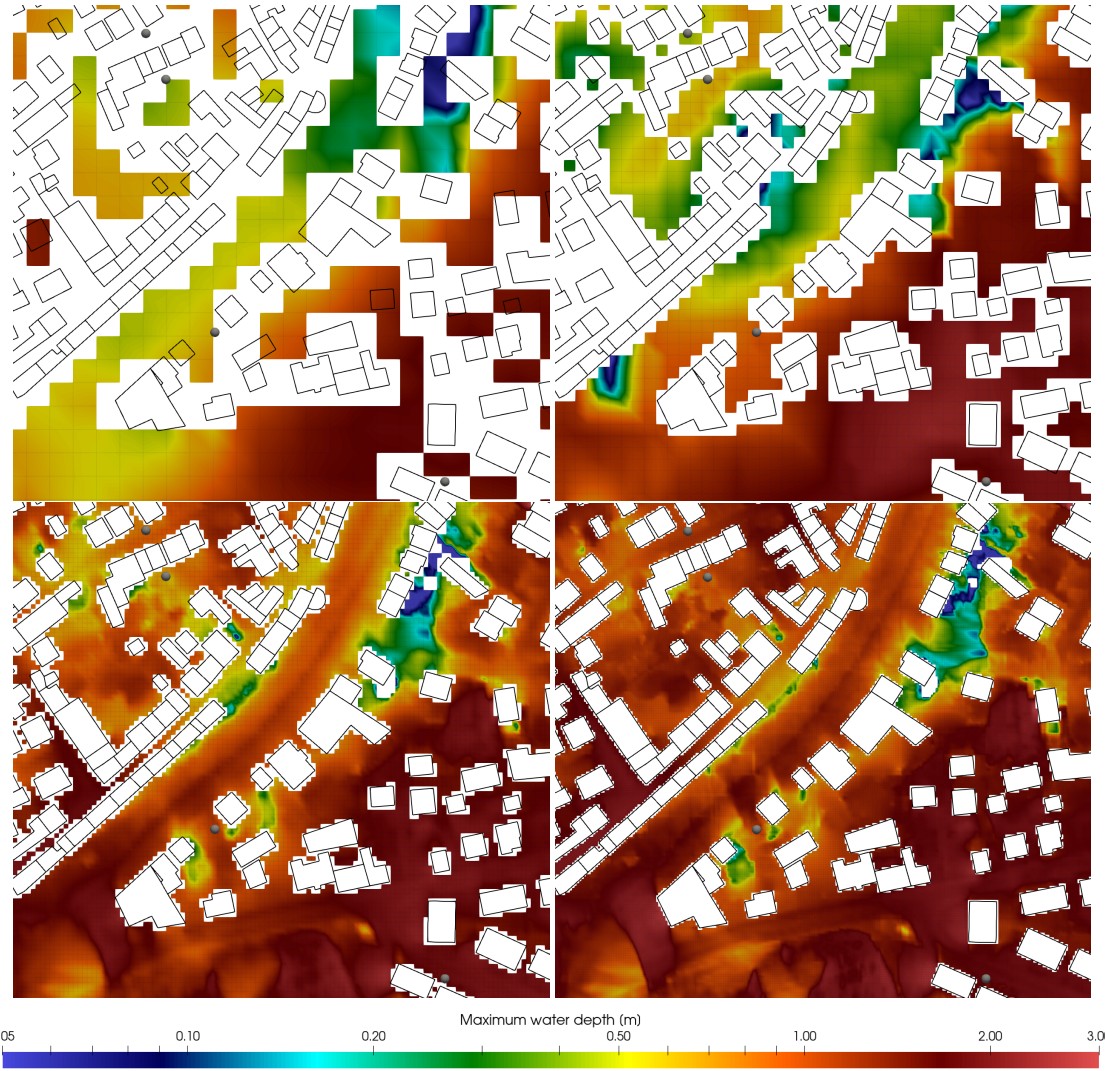

**Figure 9.** Detailed view at Ahrweiler of the maximum water depth at four different resolutions (10m, 5m, 2m, 1m from top left to bottom right) predicted by SERGHEI, together with the location of some observation points (gray dots) and the building footprints (black lines). White areas are grid cells excluded from computations as they are flagged as buildings. The figure shows how the observation points may fall in cells which are flagged as buildings at coarser resolution.

## 3.6 Flood evolution

Figure 10 shows a comparison of the evolution of flooded areas with increasing water depths for both solvers and all resolutions. Complementary, Figure 11 shows the fraction of flooded area larger than a certain depth threshold, relative to the full extent of the flood. In physical terms, Figure 10f corresponds to the very deeply flooded areas, and of course includes the main channel. This is of course a rather small fraction of the flooded area (less than $10\%$ for most of the simulations as shown in Figure 11). In contrast, Figure 10a reflects most of the flooded area (only excluding areas flooded with less than 5 cm of water). Arguably, water depths below 10 cm only reflect an inconvenience in terms of flood impact. However, depth around 50 cm already include flooded underground and ground floors in buildings, have transport potential to move unsecured objects and represent a danger to human life. A very large fraction (between $\sim 80\%$ and $\sim 90\%$ at the peak) of the flooded areas is indeed flooded with more than 50 cm of water, and up to around $40\%$ to $50\%$ of the flooded area exceeds 2 m of depth. This strongly underlines the high impact of this flood event.

In comparative terms, the flooded area evolution for all water depths shows similar behaviours, especially in terms of interpreting the results for flood impact and warning. Nonetheless, some deeper reading of the differences proves insightful.

The SERGHEI simulations show a clear trend of decreasing peak flooded areas and delayed peaks with coarser resolution. This is expected and consistent with well known hydrograph attenuation and delay due to numerical viscosity (diffusion) (Caviedes-Voullième et al., 2012).

Interestingly, for RIM2D the effect of resolution is the opposite as in SERGHEI. The local-inertia solution results in higher peak areas for coarser resolutions. Additionally, no significant delay of the peaks is observed in the RIM2D flooded area curves. The insensitivity in the timing to resolution is consistent with the behaviour of diffusive-wave (zero-inertia) formulations as discussed in subsection 3.4, and these results suggest that the local-inertia approach keeps this property. It is possible that roughness calibrations could alleviate this issue.

Comparing across solvers for the same resolution shows (i) for the coarser grids (5m and 10m) SERGHEI results in smaller flood extents than RIM2D across all depth thresholds; (ii) for the finer grids (1m and 2m) SERGHEI results in larger flood extents than RIM2D across all depth thresholds; (iii) the peak of the flooded area curves is somewhat earlier for RIM2D than for SERGHEI, for all resolutions. Observations (i) and (ii) are explained by the previous discussion on the effects of resolution on the different solvers.

Observation (iii) is consistent with the discussion in 3.4. Although the lack of convective terms in the local-inertia equation typically leads to slower wave propagation in comparison to the full shallow water equations (De Almeida and Bates, 2013; Martins et al., 2017), this is not reflected in Figure 10. It is likely that the complex dynamics of wave propagation and flood buffering in the channel and floodplains may play a more significant role than the attenuated wave propagation speeds. Moreover, as noted by De Almeida and Bates (2013), the relevance of this wave slowdown is greater for higher Froude numbers. In this event, the simulations show that most of the flow field experiences sub-critical conditions (in fact, mostly with Froude $< 0.6$). This suggests that the wave slowdown in the local-inertia solver may not be very significant except for very local areas with higher Froude numbers.

Another important aspect in the evaluation and discussion of the simulation results above is that all simulations used the same set of roughness parameters. Many studies Costabile et al. (e.g., 2023); De Almeida and Bates (e.g., 2013); Pappenberger et al. (e.g., 2005) emphasised that roughness used in surface flow solvers is not absolute, but has to be regarded as effective roughness. This means that roughness is the main calibration parameter for hydraulic models, which can compensate for effects of model formulations (solvers) and model setups (resolution) on simulation results (Caviedes-Voullième et al., 2012; Caviedes-Voullième et al., 2020; Costabile et al., 2017). With dedicated calibrations of both RIM2D and SERGHEI models for different resolutions it can be expected to reduce the differences in the model results. However, this is out-of-scope for this study, which aims at exploring the differences in simulation results caused by solvers and spatial resolution in 2D hydrodynamic models using standard roughness values, just as a modeller might do when exploring potential floods in a new setting and context (precisely what our simulation exercise was intended to represent). A comprehensive study of the sensitivity of roughness coefficients on the RIM2D local-inertia solver is expected as future work.

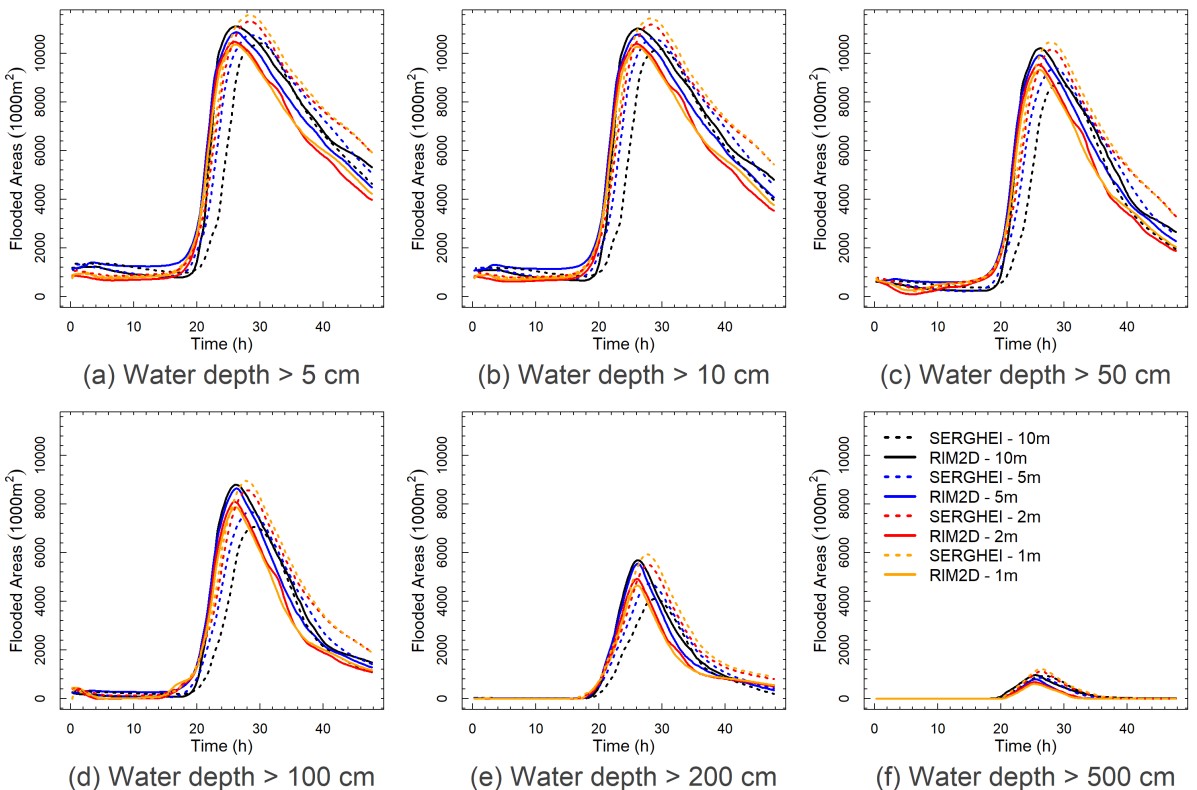

**Figure 10.** Inundation areas with water depths above 5, 10, 50, 100, 200 and 500 cm during the dx= 1, 2, 5 and 10 m simulations. The values have been measured with a 900 second temporal resolution

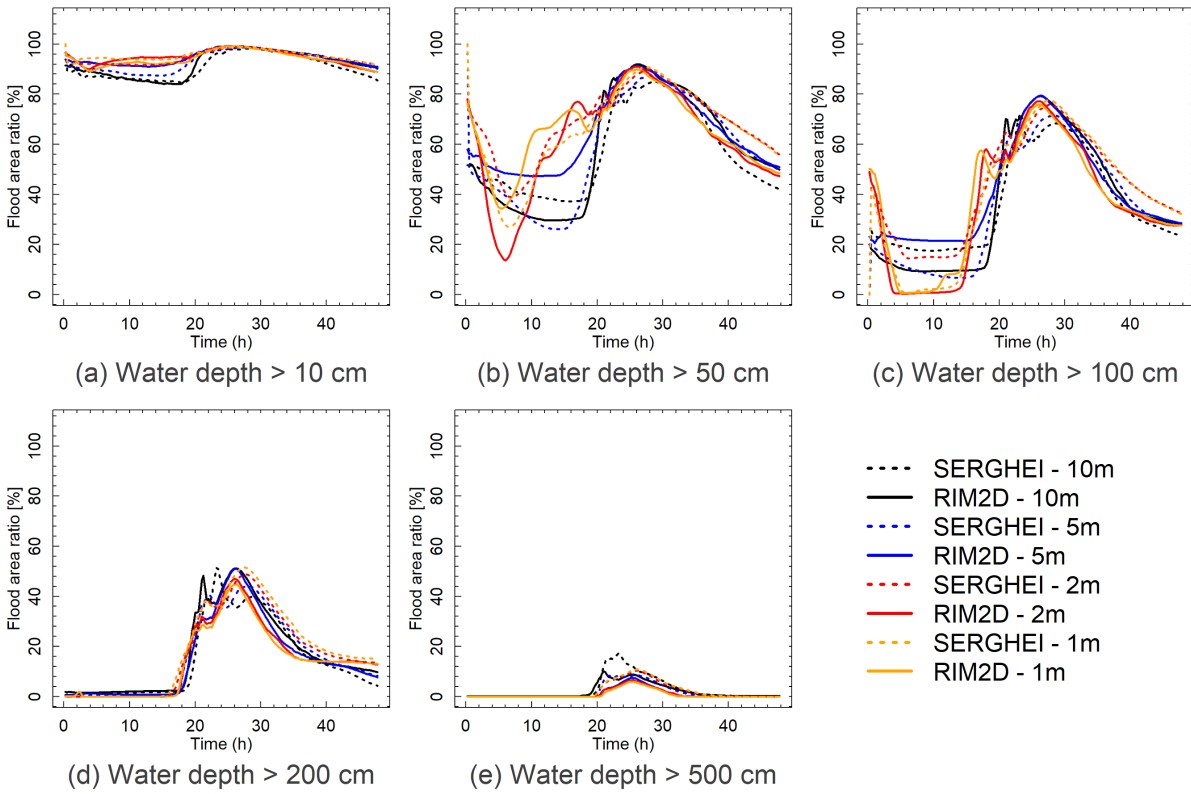

**Figure 11.** Flood area ratio of water depths above 10, 50, 100, 200 and 500 cm during the dx = 1, 2, 5 and 10 m simulations compared to the flooded area of the corresponding simulations with water depth above 5 cm . The values have been measured with a 900 second temporal resolution

## 4 Conclusions

In this study we demonstrate that the state-of-the-art in 2D surface flow modelling currently allows for significantly faster-than-real-time simulations of flash flood events, such as the July 2021 Ahr valley flood event. Evidently, runtime remains a function of the model used (in our case the local inertia solver RIM2D, and the full shallow water solver SERGHEI), the target resolution for the forecast (here between 1 and 10 m) and the computational hardware (here we used between 1 and 8 scientific grade NVIDIA A100 GPUs).

We show that for this particular event, it is currently possible to generate flash flood forecasts 304 times faster-than-real-time at 10m resolution, and 99 times faster-than-real-time at 5m resolution. Using HPC resources with SERGHEI it is possible to achieve 8.2 faster-than-real-time simulations even at 1m resolution. This holds particular significance, especially regarding the Ahr Valley floods, where the type and timing of flood warnings were pivotal in determining the extent of the casualties, together with the shortcomings of existing warning systems (Thieken et al., 2023a), including short lead times, untimely alerts, outdated or inaccurate data and inconsistent guidance (Thieken et al., 2023b). Traditionally, many areas rely on early warning

systems that communicate information primarily based on rainfall amounts and water levels or discharges at a limited number
of river gauges. However, the Ahr Valley floods highlight the potential limitations of such systems, particularly in scenarios
where detailed information on the inundation extent, expected water levels in the inundated areas and water arrival times are
essential for effective response and decision-making. The models employed in this study demonstrate the ability to simulate
water levels, inundated areas and flood propagation with a high level of detail and accuracy.

The detailed analysis of the two solvers applied to a range of different spatial model resolutions using the same set of
hydraulic roughness showed large similarities in simulation results in terms of inundation extent and depths. Some differences
were also observed in terms of timing of flood peaks and wave propagation. These differences can be explained by the different
mathematical foundations of the models (i.e., local inertial formulation vs. full shallow water equations) and the resulting
differences in simulated wave propagation and dependencies of simulation results from spatial resolution. Knowing about
these differences as laid out in the result section helps in selecting the appropriate model and spatial resolution for the problem
to be studied, as well as for interpreting the results. This mainly accounts for flood propagation speed and flow velocities, and
less for simulated water depths and flood extent, which is traditionally the main concern in flood forecasts. From a practical
point of view of deciding on model complexity for these type of events, our results place the comparative behaviour of the
local-inertia approximation relative to the full shallow water equations in the expected ranges, with flood extension being
not very sensitive to the selected model, whereas hydrodynamic fields are more sensitive (Caviedes-Voullième et al., 2020;
Costabile et al., 2019).

Another relevant practical insight of this study is the value of resolution in flood simulation for early warning. Although the
skill metrics are in general terms acceptable the different resolutions, higher resolution still generally improves results. Broadly,
there seems to be some significant change in the behaviours below 5m resolution, which may be attributable to better resolved
topography or buildings. Arguably, 10m resolution (although providing good model skill) may be too coarse to provide accurate
details in urban areas, simply because relevant features are not resolved, and with the computational efficiency shown here, it
is absolutely feasible to move to higher resolutions to avoid this risk.

Considering that the presented models are not calibrated, and that compared to the inherent uncertainties in flood forecast
chains originating from uncertainties in rainfall forecasts and hydrological modelling, the uncertainties introduced by the
choice of the hydraulic model and spatial resolution is comparatively low (Apel et al., 2008; Sampson et al., 2014). Thus,
the choice of the hydraulic model can be rather based on the required simulation runtimes, spatial resolution and available
computational resources, rather then on the specific hydraulic properties of a particular solver. For the presented test case in
the Ahr valley, spatial resolutions of 5m and even 10m would yield forecasts sufficient for actionable flood response, with
both solvers providing valid simulation results with simulation runtimes short enough for use in operational flood forecasts.
Calibrated models are of course expected to perform even better.

In summary, the key outcome of this work is a proof-of-concept that this technology is mature enough to be up taken into
early warning systems, ensuring sufficient lead time, and providing far more informative and actionable results than traditional
flood early warning systems. High resolution and time resolved depth and velocity fields provide a far better picture of flood
severity and allow for additional analytics to derive impact metrics, and are much more easily intrepretable by the general

public, managers, and emergency responders compared to warnings based on communicating, for example, rainfall amounts
(Thieken et al., 2023b). The key next steps involve implementing these solvers into workflows which more broadly cover the
data and modelling chains for early warning systems.

*Code and data availability.* SERGHEI is available through GitLab, at https://gitlab.com/serghei-model/serghei, under a 3-clause BSD license. Simulations were carried out with SERGHEI v1.1. RIM2D is available at https://git.gfz-potsdam.de/hydro/rfm/rim2d. RIM2D is available for scientific use under the EUPL1.2 license. Access is granted upon request. The simulations were performed with version 0.2.

The digital terrain models used here are published produced in the spatial database of the German Federal Agency for Cartography and Geodesy (BKG), available at https://gdz.bkg.bund.de/index.php/default/digitale-geodaten/digitale-gelandemodelle.html.

The OSM building shape files used in this research can be freely obtained from https://download.geofabrik.de/europe/germany.html. The land cover raster, which was used to assign roughness values to the simulation domain, is openly accessible at https://www.mundialis.de/en/germany-2020-land-cover-based-on-sentinel-2-data.

Flood extent data was obtained from the UFZ data investigation portal via DOI: 10.48758/ufz.14607.

*Author contributions.* DCV & SKBG: Conceptualisation, Methodology, Investigation, Software, Formal Analysis, Visualisation, Writing. HA: Software, Writing

*Competing interests.* The authors declare no conflicts of interests.

*Acknowledgements.* The authors gratefully acknowledge the Earth System Modelling Project (ESM) for supporting this work by providing
computing time on the ESM partition of the JUWELS supercomputer at the Jülich Supercomputing Centre (JSC) through the compute time project *Runoff Generation and Surface Hydrodynamics across Scales with the SERGHEI model* (RUGSHAS), Project Numbers 26702 and 29000. This research was performed within the frame of the DIRECTED project (https://directedproject.eu/). Funding of the DIRECTED project within the European Union's Horizon Europe – the Framework Programme for Research and Innovation (grant agreement No. 101073978, HORIZON-CL3-2021-DRS-01) is gratefully acknowledged.

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
