# Peer review of "Are 2D shallow water solvers fast enough for early flood warning? A comparative assessment on the 2021 Ahr valley flood event"

_Natural Hazards and Earth System Sciences, 2024_

## Author Response (AR1)

**Author's Response**

**Reviewer 1**

Dear colleagues, thank you very much for an interesting paper. I enjoyed reading the paper and I would like to send you my comments and remarks:

Thanks for the kind words and nice feedback.

1. The aspect of early warning is mentioned in the title but not really discussed in the paper. First of all, fast simulations are necessary but more aspects are required to make a code suitable for Early Warning. Is a code able to include weather forecasts? Is a HPC-computer required or can simulations be run on normal computers. Are code and results accessible to those institutions who are responsible for flood forecasts and flood warning. I have the impression that both codes and especially SERGHEI is more an academic code and not suitable for practical applications. Therefore, the title of the paper is misleading and the core of the paper is more a comparison of different codes and different resolutions.

**REPLY**: We do agree with this assessment, also pointed out by Reviewer 2. We did not intend to disregard the complexity of everything else required for early warning systems, and we do see how the title is in that sense misleading. Thanks for pointing it out. We have of course changed the title to convey what this paper is really about, within the context of improving early flood warning.

2. A second weak point of the paper is the missing calibration. I had expected a calibration on the basis of the 2016 event in the AHR valley or the 2021 event. Even gauges were distroyed during the 2021 event, data are available which can be used for verification of the results. Just a comparison between two codes is not sufficient.

**REPLY**: We would argue here two points: - we purposely do not want to do any calibration. In this sense, the exercise aims to perform blind simulations based on the minimal data necessary (and available!) to run such simulations. The reasoning behind this is precisely to assess the capabilities, and arguably the advantage that these models have in performing blind simulations. We have included some remarks about this in the manuscript, as it has become clear that we did not convey these intentions and reasoning in the original manuscript. - we do not only compare the two codes but actually compare to field data and gauge data. This shows in in Figure 2, Table 4, Figure 7 (which is new, see below) and Figure 8 and we accordingly draw conclusions of the model skill. We do of course pursue further comparisons to understand the differences between the models. What is indeed true, is we do not take on a calibration study using this data, following the argument provided above.

3. Figures can not be used for interpretation of results. As an example, it is difficult to interpret Figures 4 and 5. I had at least set a clear change in colours at 0.0 in both figures. you have chosen a continous change of

colours. Therefore, interpretation of results is not possible. Try to use less colours instead of two many colours.

**REPLY**: We don't fully understand what the reviewer means by "Figures cannot be used for interpretation of results". We interpret that perhaps the point is that interpretation of the figures is only semi-quantitative. Figures 4 and 5 provide an overview of the differences between the two models. The allow mostly for qualitative and broad intepretations which can allow to understand implications on modelling choices (i.e., model formulation, resolution). The colour scale is deliberate, to span negative and positive differences, and we purposely use a continuous scale to avoid biasing the analysis by binning into more discrete color categories. For example, we could choose to color the range of differences smaller than -2 in the same color, and we would lose information in the reach around Rech at dx=10. We would argue that the color scale is both broad and resolved enough to broadly read into the differences, albeit, we agree, it requires some time to digest. Nevertheless, we do not expect the reader to draw their own conclusions blindly either, but we of course highlight what we believe are the key features of the differences in the simulations, both in terms of predicting depth and predicting the timing.

4. the lag difference is not very well explained. Please describe more accurately, what you have analysed.

**REPLY**: We have add a bit more details in how the lag is computed, which should make it more accesible to follow the arguments.

5. I see a number of coloured points in Figure 7. No interpretation is possible.

**REPLY**: Figure 7 is now Figure 8. Thanks for pointing out that the interpreation of this figure was unclear. The intention of the figure is to provide a spatial overview of the differences between simulations and field observations. The intended interpretation relates here to WHERE results deteriorate and where the do not. However, we can see that the quantitative interpretation is difficult. Consquently, we have added a new Figure 7 (and Figure 8 is now the old Fig 7) which shows a scatter plot of simulated vs observed depths at the location of the field observations. This provides a more quantitative and structural understanding of the differences, although it loses the spatial aspect (which is still what Fig 8 conveys). The two figures nicely complement each other and the discussion.

6. A description is required in Figure 8. the legend is too small. Changes are not obvious.

**REPLY**: This is now Figure 9 in the revised manuscript. We have moved the legend outside of the plots for clarity. We have also included a sentence on what is the main focus of the figure in the caption.

7. What is the main contribution of this paper? It is more a comparison of different spatial resolutions without any comparison to field measurements.

Therefore, I recommend significant improvements of the paper. How and to which extent can your results be used by practice?

**REPLY**: We hope that the main contribution of the paper is now more clear following the new title: it is about showing that it is indeed technically possible to do fast and accurate simulations with these solvers within the lead times requried for early warning. Moreover, it is possible to do with off-the-shelf datasets, withouth calibration, with a reasonably good accuracy.

We would remind the reviewer that we did compare to field measurements, as previously argued.

In terms of applicability to practice, we basically claim that the technology is ripe and available. This should pave the way into early flood warning systems to uptake these type of models as core tools in the workflow. The applicability is that these solvers will deliver simulations within the lead times required, with far more information, physicallity and resolution than, for example, conceptual models, parametrised flood models, or 1D models.

**Reviewer 2**

General Comment

This paper addresses a topic of great interest, focusing on the analysis of predictive performance and computational efficiency of 2D propagation models for use in early warning systems. This topic is highly relevant. I believe that the overall approach proposed by the authors represents the most up-to-date methodology currently available, given the application of multi-GPU and HPC computing capabilities. Moreover, the comparative analysis of two 2D models with different conceptual approaches (fully dynamic 2D SWE equations and local inertial approximation) in a complex case study, as examined here, adds significant value to this research. While there are other works demonstrating how hydrodynamic approaches with HPC technology can effectively be used as modules within more articulated early warning systems, I believe this work has its own specificities and represents an original contribution to the literature. The paper is well-written and easy to read and follow. The methodological approach is clear, and the analysis of the results is well-discussed and complete. Though the conclusions are, in part, dependent on the specific test case analysed here, there are generalizable findings and suggestions. Therefore, I expect that this paper may have a large impact on the related literature.

**REPLY**: Thanks for the positive feedback.

I have only a few minor points that the authors could consider to further enhance the clarity of their work.

Specific Comments

1- While I fully share the authors' perspective, I believe that the title could be enhanced. The paper's organization primarily emphasizes the comparison between

the predictive capabilities of the two models (SERGHEI and RIM2D). Both models are employed as traditional propagation models rather than hydrodynamic-based rainfall/runoff approaches, which are more aligned with early warning models. These simulations can be activated by rainfall predictions/measurements, as the authors are aware. I'm concerned that the title may create expectations that are not fully met in the paper, as there is no direct link with the meteorological/hydrological components of early warning systems.

**REPLY**: Yes, we fully see the point, also raised by Reviewer 1. We have changed the title accordingly, putting the emphasis where it should be, but mainting the outlook towards early warning systems. Indeed, we do not wish to claim that we present an early warning system, but rather that the numerical solvers are ready to be uptaken by early warning systems. We think the new title conveys this.

2- The paper presents the governing equations of the SERGHEI model but does not include those of RIM2D. I would suggest either adding the governing equations of RIM2D for a more comprehensive understanding of both models or removing the SERGHEI equations. Presenting only one set of equations might give the impression of partial coverage.

**REPLY**: Thanks for pointing out this oversight. We agree and have included equations which clearly show the local inertia simplification.

3- In Section 2.2 (Study Case), I would suggest explicitly mentioning the available data for the reconstruction of the event. While some information is scattered throughout the paper, consolidating all of it here could enhance clarity and benefit the readers.

**REPLY**: This is a very good idea. Indeed, we were introducing the observational data at the point in the paper where it is used, but this was indeed a bit unclear. It is now solved.

4- Section 2.3. Lines 129-132 are not completely clear to me. Could you provide further clarification or rephrase this section?

**REPLY**: We meant to say that we applied both codes "as is", on the available hardware without undergoing any kind of computational optimisation strategy. Perhaps what was missing (and now was added) is what we meant by the optimisation. We mean things, for example, like testing optimal domain decomposition strategies, or playing with compilation and compiler optimisation flags to squeeze out additional performance. Such approach potentially can reduce runtimes even further. However these optimisations are not generalisable across cases, software stacks and hardware, so we don't see the value of doing this in an exploratory study, but recognise that under operational conditions, this can and should be done.

5- Line 145: What about the values of qx and qy at the inflow cells? If supercritical flow conditions occur, you should also set these values, at least for the fully dynamic model.

**REPLY**: This is a very interesting question. There are two competing interests in the modelling choices we took here: optimality and comparability. The optimal inflow boundary condition should be a (qx,qy) hydrograph, which can be constructed by assuming inflow normal to the cross-section and relying on an inflow hydrograph. However, the comparability of our exercise would be reduced, since RIM2D does not implement such a boundary condition. As you well point out, an inflow hydrograph would be the optimal choice for SERGHEI. We did carry out some small comparisons of inflow vs stage hydrograph boundary conditions with SERGHEI, and found minor local differences. This may be because of the encroached valley at the inlet, which leads to rather high water depths, making this the dominant factor.

Results and discussion 6- I suggest considering moving Section 3.2 to the beginning or, even better, to the end of the analysis of the predictive capabilities of the models. In the current version of the manuscript, it is mixed with the runtime analysis, which may create some confusion for the reader.

**REPLY**: Thanks for this suggestion. We have moved it to the start of our results section, since the runtime is a key point in this work, especially in line with the new title.

7- Did you check for the presence of artificial depressions generated when deriving DGM10? This may impact the results and introduce some confusion in the interpretation, especially when considering the time to maximum depth.

**REPLY**: No. We took the 10m product that is provided by the German Federal Agency for Cartography and Geodesy (BKG). We assume this to be properly treated, since we purposely want to use available products and data sources. If aritifical depressions do exist in such DTM, our argument here is that it is simply a limitation of the datasets. Of course, we agree that one can to some extent correct such artifacts, but we refrain from carefully crafting a model domain, since we want to keep the exercise in the realm of emergency early warning, in which such careful crafting may not be possible.

8- Section 3.4: While the results are influenced by the choice of Manning coefficients reported in Table 1, it is commendable that the authors obtained good results through a "blind" approach. However, I believe the impact of Manning's values on the two models may differ. I wonder how the results obtained by the models could be affected by variations in Manning coefficients. This could roughly quantify the uncertainty in model predictions. It is not mandatory, but it would be interesting to include this information.

**REPLY**: We would reply to this with three points: - as you say, we have purposely opted for a blind approach to place the exercise in the context of forecasting a flood, in an uncalibrated fashion, possibly for a previously unseen event. We have realised though, thanks to this comment and a related one from R1, that we did not state these intentions and the rationale behind it. We now have done so. - Previous work shown between zero-inertia and dyanamic SWE solvers (including your work), and it seems reasonable that something similar

will happen with local-inertia. Our results in this paper indeed suggest that the impact of roughness values (and consequently their calibration) will be different in both formulations. This insight can be extracted by the fact that keeping the roughness constant across resolutions and solvers, there are differences in the wave propagation speeds, which are at least partially attributable to roughness. SERGHEI tends to get timings better than RIM2D with this blind roughness parametrisation. Experience with RIM2D shows that a suitable roughness calibration (which is resolutioin dependent) can improve this. The consequence of this is precisely that the calibration of Manning values has likely more relevance for a local-inertia solver. This is reasonable, since the simplifications in the local-inertia solver give somewhat of a higher weight to the friction source term. - we fully agree that a sensitivity and/or calibration study would provide further insights. A manuscript specifically targeting this for RIM2D is submitted to NHESS, thus we choose to not cover that topic in this particular manuscript, but rather keep the focus on our core message. Nevertheless, we have included some of the arguments in this reply in the text.

9- Section 3.5. I agree with your considerations on the effect of buildings (lines 276-277), essentially due to the type of grid used. Moreover, the area available for flood expansion varies with the model resolution, influencing water level and velocities. This could raise some questions regarding the reliability of 10m resolution in urban areas. Perhaps some comment on this is needed. I also wonder if it would be appropriate to model buildings with increased roughness to avoid this effect when using coarser resolution

**REPLY**: We agree that 10m resolution may be to coarse to cope with urbanised areas since it is most likely that building geometry is not well captured. Our results suggest that this is likely particularly when evaluating local dynamics, although not necessarily for the broad view of the flood. We have not undertaken a detailed analysis of specific regions in the domain (e.g., we only focused on the region shown in Figure 9), but indeed inspection of results does show some artifacts at 10m resolution and local underestimation of maximum depth. We have included some additional comments on this. However, including the buildings even at 10m resolution provides more realistic flood propagation in built-up areas compared to an urban porosity approach, where buildings are not considered as obstacles, but by increased roughness on the building footprints. Even with high roughness on building footprints the flow will not be be diverted around buildings, but routed over the building footprints at slower speed. While this is acceptable for simulating large scale flood without a special focus on urba areas, this is clearly a disadvantage when flood dynamcis in the built-up area is of importance/interest. Moreover, the raster resolution problem applies for the urban porosity appraoch just as for the presented method. And if the built-up area as a whole is presented by higher roughnes without considering the building footprints explcitly, the urban flow dynamics cannot be simuated at all.

10- Some Figures are not very easy to read and interpret. In particular, Figures 7 and 8 could be improved and the font size of Figure 10 should be increased.

**REPLY**: We have introduced a new Figure 7 which now complements Figure 8 (old Figure 7). Figure 7 provides a more straightforward quantitative comparison of simulations and observations, whereas Figure 8 intends to show the spatial distribution of the same data. Figure 9 (old Figure 8) now has a new larger legend. We have implemented improvements in Fig 10 (now 11).

11- The conclusions are relevant and effective. However, given the significant emphasis on model comparison, practical recommendations on the use of these models considering their benefits and limitations would be expected. Perhaps you could discuss this aspect.

**REPLY**: We have provided some additional outlook in terms of benefits/limitations and potential applicability.

I am fully supportive of the publication after these minor revisions and would like to commend the authors for their fascinating research.

**REPLY**: Thanks again for the positive feedback.